# Effect of Strategic Supplementation of Dietary By-Pass Linseed Oil on Fertility and Milk Quality in Sarda Ewes

**DOI:** 10.3390/ani13020280

**Published:** 2023-01-13

**Authors:** Ignacio Contreras-Solís, Cristian Porcu, Francesca D. Sotgiu, Fabrizio Chessa, Valeria Pasciu, Maria Dattena, Marco Caredda, José Alfonso Abecia, Giovanni Molle, Fiammetta Berlinguer

**Affiliations:** 1Veterinary Medicine Department, Sassari University, 07100 Sassari, Italy; 2Department of Animal Science, AGRIS Sardegna, Loc. Bonassai, 07100 Sassari, Italy; 3Veterinary Faculty, Zaragoza University, 50013 Zaragoza, Spain

**Keywords:** linseed oil, metabolic status, milk quality, polyunsaturated fatty acids, fertility, pregnancy, dairy ewes

## Abstract

**Simple Summary:**

Polyunsaturated fatty acids (PUFAs) are well known for their beneficial role in different body systems and their function in mammals. The use of PUFA-ω3 as a feed additive has been investigated with the aim of improving the reproductive and productive performance of different livestock species, but their effects on dairy sheep are scarcely documented. The present study demonstrated that in milking ewes, dietary supplementation of bypass linseed oil (LO; rich in alpha-linolenic acid/ALA) modified the lipidic metabolic status, increased milk PUFA composition, and the size of luteal tissue at the onset of maternal recognition of pregnancy (on day 11 after insemination). The present study showed that the use of a bypass LO supplementation during the periconceptional period did not increase the reproductive indexes in ewes. Nevertheless, dietary bypass LO modified the circulating concentrations of lipid metabolites and the milk fatty acid profile, demonstrating its absorption and utilization by body tissues.

**Abstract:**

The aim of the present study was to assess whether the strategic supplementation of bypass LO can enhance reproductive indexes—fertility, lambing rate, and prolificacy—in dairy Sarda ewes at the end of lactation. To assess whether LO supplementation leads to the adsorptions of PUFAs and their subsequent utilization by the body tissues, milk composition and fatty acid content were analyzed. Forty-eight ewes were assigned to the following groups: the control group (CT; *N* = 24), fed with a control diet without LO; and the treatment group (LO; *N* = 24), fed with a diet supplemented with LO (10.8 g/ewe/day). Both diets had similar crude protein and energy levels and were offered for 38 days (−21 to +17 days after artificial insemination). The trial included an adaptation period (7 days) followed by a regular supplementation (31 days) period. Estrus synchronization was induced in all the ewes using an intravaginal sponge and equine chorionic gonadotropin. Fifty-five hours after pessaries withdrawal, all ewes were inseminated using the cervical route and fresh semen. Cholesterol (*p* < 0.01), high-density lipoprotein (*p* < 0.001), and triglyceride (*p* < 0.05) levels in plasma were higher in the LO group. Plasmatic levels of non-esterified fatty acids were lower in the LO group after the end of the supplementation period (*p* < 0.05). Milk unsaturated fatty acids (UFAs), monounsaturated fatty acids (MUFAs), total polyunsaturated fatty acids (PUFAs), PUFAs omega 3 (PUFAs-ω3) and 6 (PUFAs-ω6), and trans fatty acids were higher in the LO group (*p* < 0.001), while saturated fatty acids (SFAs) were higher in the CT group during the supplementation period (*p* < 0.001). Three days after the end of the supplementation period, the content of milk UFAs (*p* < 0.05), PUFAs (*p* < 0.001), MUFAs, and PUFAs-ω6 (*p* < 0.01) were still higher in the LO group. whereas SFA was higher in the CT group (*p* < 0.01). There was no difference between groups in terms of ovulation rate, progesterone levels in plasma, fertility rate, prolificacy, and total reproductive wastage. However, the total area of luteal tissue was higher in the LO group (*p* < 0.01). Results obtained demonstrated that LO supplementation exerts a positive role in corpus luteum size at the onset of the peri-implantation period in Sarda dairy ewes. Additionally, the results obtained in the present study showed that the use of dietary bypass LO affects lipid metabolites in plasma and milk fatty acid profiles, demonstrating the ALA uptake by body tissues.

## 1. Introduction

During the last decades, the efforts to develop and design green and clean strategies to increase the productivity and sustainably of the livestock sector have markedly increased. Sheep production is an important livestock sector in Europe and other developing countries. Thus, in-depth research studies must be led to design and apply these strategies in this species to guarantee animal welfare, offer healthy products (e.g., milk and meat) for human consumption, and reduce the flock’s operative cost.

In the Mediterranean basin, dairy sheep production usually implies one lambing per ewe. Adult ewes are usually mated at the end of the seasonal anoestrus period (May–June), while they are passing from mid to late lactation (5–6 months of milking). During this period, the pasture usually turns to its productive phase, resulting in a diminishing intake of nutrients and lower nutritive value [1]. Thus, appropriate nutritional supplementation is pivotal to sustain productive performances, as it should be formulated to meet the requirements of both the reproductive and mammary systems to increase the fertility of the flock while sustaining milk production.

Long-chain polyunsaturated fatty acids omega 3 (PUFAs-ω3) and 6 (PUFAs-ω6) are well-known for their beneficial role in different body systems in ruminants. These PUFAs are considered essential because they cannot be synthesized by body tissues in mammals [2]. Alpha-linolenic acid (ALA) has been identified as the most important PUFA-ω3 from plant origin [2,3,4], due to its effects on reproductive indexes and milk quality in dairy cattle [3,5]. The ALA content ranks from 0.6 to 3.2% dry matter in most fresh forages [6]. However, its availability depends on plant maturity, seasonal changes, and the ruminal biohydrogenation process [7]. Flaxseed/linseed oil (LO) is one of the most valuable sources of ALA used as feedstuff in large and small ruminants [3,8]. Its use as a nutraceutical/feed additive is achieved using a protected bypass form to avoid ruminal degradation (biohydrogenation) and guarantee its bioavailability/absorption at the intestinal level [9]. 

Previous studies have attributed LO (rich in ALA) the capability to increase follicular development and ovulatory rate, as well as fertility rate in dairy cows [10]. In addition, LO has shown a luteotropic effect [11]. However, other studies did not confirm these findings in large and small ruminants, suggesting that additional studies were necessary [3,11]. 

The use of LO supplementation is thus considered a promising nutraceutical tool to enhance reproductive performances in ruminants with particular reference to dairy cattle [3,11]. Starting from these premises, the aim of the present study was to assess whether the strategic supplementation of bypass LO can enhance reproductive indexes—fertility, lambing rate, and prolificacy—in dairy Sarda ewes at the end of lactation. To assess whether LO supplementation leads to the adsorptions of PUFAs and their subsequent utilization by the body tissues, milk composition and fatty acid content were analyzed.

## 2. Materials and Methods

The study was carried out from June to July at the “Bonassai” experimental farm belonging to Agris Sardegna, the agricultural research agency of Sardinia, Italy (40°40′24″ North and 8°21′59.4″ East). The experimental procedures were performed according to the rules of the Ethic Committee for Animal Lab Management from the Italian Health Ministry (Authorisation n° 757/2020-PR/ Protocol-E8652.2). The experiment lasted 38 days and was divided into three phases, starting with (1) the adaptation period (PRE period; from day −21 to −15 before fixed artificial insemination, FAI (day 0). During this period, the ewes were familiarized with a new environment and dietary regimens were gradually changed to offer the final diet composition for supplemented and non-supplemented ewes (Table 1). (2) In period 1 (Per1; from day −14 to −1), supplementation of the final diet compositions was before FAI. (3) In period 2 (Per2; from days 0 to +17), supplementation of the final diet compositions was after FAI. Carry-over effects were studied in the post-treatment period (POST; day ≥ +18).

## 3. Animals

Forty-eight Sarda milking ewes (4.23 ± 0.10 years old) were selected from the farm flock at the very onset of the adaptation period and blocked by age, body weight (BW, kg), body condition score (BCS), milk yield (MY, g/d), and days in milk (DIM, d), and divided into two homogeneous groups. They were allocated in two different pens in the same sheep shed and assigned to the following experimental treatments: the LO-supplemented group (LO; *N* = 24), which was fed with a diet supplemented with bypass ALA (using bypass LO; SILA™; Verona, Italy; 18% ALA); and the control group (CT; *n* = 24), which was fed with a control diet without LO. 

The ewes were machine-milked once daily between 7:00 and 8:00 a.m.

At the beginning of the experiment (PRE period; on day −21), the groups were undifferentiated for age (CT = 4.35 ± 0.13 years old and LO = 4.10 ± 0.15 years old; *p* = 0.069), BW (CT = 50.62 ± 0.90 kg; LO = 50.64 ± 1.15 kg; *p* = 0.989), BCS (CT = 2.83 ± 0.05; LO = 2.87 ± 0.06; *p* = 0.695), MY (CT = 920.83 ± 71.32 g; LO = 908.33 ± 61.67 g; *p* = 0.895), and DIM (CT = 203.67 ± 1.62 days; LO = 202.38 ± 1.65 days; *p* = 0.579). 

## 4. Animal Feeding

Individual concentrate daily supplementation consisted of a mix of pelleted concentrate (300 g) and barley (200 g) in CT ewes, and a mix of pelleted concentrate (200 g), barley (120 g) and bypass LO (60 g) in LO ewes. Concentrate mixes were offered in the milking parlor during milking. In addition, all ewes were group-fed daily in separate troughs with 1000 g of ryegrass (*Lolium multiflorum*) hay and 600 g of lucerne (*Medicago sativa*) hay per animal. Both diets were designed to supply similar crude protein and energy levels (isoproteic and isoenergetic diets, (Table 1). 

## 5. Animal Reproduction

From day 14 to day 2 (Per1), all ewes were treated with progestogen (20 mg of fluorogestone acetate, FGA, Chronogest; MSD, France) using intravaginal pessaries left in situ. On day 2, equine chorionic gonadotropin (eCG) was administered (350 IU/ewe; Folligon, MSD, The Netherland) at pessary removal. Fixed-time artificial insemination (FAI) was performed fifty-five hours after pessary removal. All inseminations were carried out by the same experienced operator using the cervical route, using fresh cooled semen from 4 fertile rams. The same number of ewes/group was assigned to each ram. 

Fresh semen was obtained from ejaculates collected by an artificial vagina. The ejaculates were immediately placed in a water bath at 30 °C for their evaluation. Total motility, progressive motility, and concentration were evaluated under phase-contrast microscopy at 38 °C on a hot plate by using computer-assisted sperm analysis (CASA) with Ceros II systems evaluation by Hamilton–Thorne. Only ejaculates with total motility ≥ 60%, progressive motility ≥ 30%, and a spermatozoa concentration ≥ 3 × 10^9^/mL were used. Then, the ejaculates from each ram were diluted using an Ovixcell extender (IMV technologies, L’Aingle, France) to obtain a concentration of 1.6 × 10^9^ spermatozoa/mL at 38 °C. After dilution, semen was placed in cold storage at 15 °C for 30 min. Since then, the ejaculates were loaded into 0.25 mL plastic straws obtaining a final insemination dose of 400 million spermatozoa. Then, AI was carried out within 7 h after semen collection [12].

## 6. Measurements

### 6.1. Body Weight and Body Condition Score

Body weight (BW) and body condition score (BCS) were measured before feed intake on days −21 (PRE period), −2 (Per1), and 18 (POST period). Body weight was measured using an electronic balance and BCS was measured by a skilled evaluator using a 5-unit scale (1, emaciate to 5, obese), according to Russel et al. [13]. 

### 6.2. Diet Intake

Concentrate for each experimental group was offered individually during milking and orts were weighed to calculate concentrate individual intake. Hay and lucerne fed to each group were weighted before feeding and orts were measured 24 h later to calculate the daily intake per group. Individual average intakes of hay and lucerne were calculated by dividing the daily intake per group by the number of ewes assigned to each experimental group. The mineral supplement was offered to each group using mineral blocks. Blocks were weighted each week in each pen and daily individual mineral intake was calculated for each group as follows: total block’s weight pre-post difference/(number of days × number of ewes). 

### 6.3. Ultrasonographic Scanning

Transrectal ultrasonographic scanning was performed to determine the presence, number, and size of corpora lutea (CL) at day 11 post-insemination (Per2). Additionally, it was used to determine the pregnancy at day 28 post-insemination (POST). Ultrasound scanning was performed, as previously described [14], by the same operator, who was blind to group composition. An ultrasonographic machine (Model ProSound 2V, HITACHI-ALOKA Medical Ltd., Mitaka, Japan) fitted with a 7.5 MHz linear array probe (82 mm prostate transducer UST-660-7.5, Aloka Co., Mitaka, Japan) was used. Each ovary was scanned several times from different angles to determine the presence and the number of all CLs. Then, the large image was frozen and the CL area was measured using electronic calipers. Pregnancy diagnosis was determined by the presence of enlargement of uterine horns and an embryo heartbeat [15].

### 6.4. Milk Yield and Milk and Blood Sampling

Milk yield was measured on day −21 (PRE), −1 (Per1), +12 (Per2), and +20 (POST). Additionally, individual milk samples (10 mL) were collected at the same time milk production was recorded. Samples were stored at 4 °C and analyzed within 4 h after sampling.

Blood samples were collected at fasting (0700 h), right before the milking procedures, on days −21 (PRE), −1 (Per1), +11 (Per2), and +18 (POST). Blood samples were collected using a vacuum collection tube with lithium heparin (VACUTEST KIMA srl—Via dell’Industria 12-35020 Arzergrande-PD-Italy) for the blood metabolites assay. On days 11 (Per1) and 18 (POST), a second blood sample was collected using 10 mL vacuum collection tubes with spray-coated K3 EDTA (VACUTEST KIMA srl—Via dell’Industria 12-35020 Arzergrande-PD-Italy) for the progesterone assay. Immediately after recovery, blood samples were cooled at 4 °C and centrifuged at 1500× *g* for 15 min. Plasma was removed and stored at −20 °C until assayed.

## 7. Chemical Analyses

### 7.1. Progesterone

Progesterone analyses were performed using an Enzyme-Linked Immunosorbent Assay kit (DiaMetra S.R.L., Perugia, Italy), designed to quantify P_4_ in humans and validated for ovine species [15]. Sensibility, intra-, and inter-assay CV were 0.05 ng/mL, 4, and 9.3%, respectively. 

### 7.2. Blood Metabolites

Samples were analyzed to determine cholesterol (CHOL), triglycerides (TRY), low-density lipoprotein (LDL), high-density lipoprotein (HDL), non-esterified fatty acids (NEFAs), protein (PROT), albumin (ALB), and urea (UREA) levels in the plasma. All measurements were performed using commercial kits (Real Time Diagnostic Systems kits) and a BS-200 Mindray (Adaltis, Milan, Italy) clinical chemistry analyzer (Mindray Medical S.R.L). Serum I Normal (Wako) and Serum II Abnormal (Wako) were used as a multi-control for each measured parameter. Sensibility from different metabolites was 3 mg/dL (for CHOL, and TRY), 2 mg/dL (for LDL and HDL), 0.01 mmol/L, 0.27 g/dL, 0.11 g/dL and 4.9 mg/dL (for NEFA, PROT, ALB, and UREA, respectively). Intra-assay CVs for CHOL, TRY, LDL, HDL, NEFA, PROT, ALB, and UREA were 1.3, 0.99, 1.0, 1.55, 1.07, 2.90, 1.88, and 1.63%, respectively. Inter-assay CVs for CHOL, TRY, LDL, HDL, NEFA, PROT, ALB, and UREA were 1.24, 1.24, 1.6, 1.75, 1.15, 1.92, 1.52, and 1.72%, respectively.

### 7.3. Milk Analyses

Milk samples were assayed to determine fat, protein (N × 6.38), casein, and lactose using the Fourier-transformed infrared (FTIR) method (Milkoscan FT + Foss Electric, Hillerød, Denmark). Quantification of fatty acids (C4:0, C6:0, C8:0, C10:0, C12:0, C14:0, C16:0, C18:0, C18:1 9c, C18:2 9c 12c, C18:2 9c 11t, and C18:3 9c 12c 15c) and of the categories of fatty acids in milk (saturated fatty acids (SFAs), unsaturated fatty acids (UFAs), monounsaturated fatty acids (MUFAs), polyunsaturated fatty acids (PUFAs), omega-6 fatty acids (PUFAs-ω6), omega-3 fatty acids (PUFAs-ω3) and trans fatty acids (trans FAs) was carried out by means of the FTIR method using prediction models previously built and validated by Caredda et al. [16]. Somatic cell count was determined using the flow cytometry method (Fossomatic 5000, Foss Electric, Hillerød, Denmark).

## 8. Statistical Analyses

Statistical assumptions (normality and homoscedasticity tests) were tested before further data analyses. Non-normal and heteroscedasticity data were analyzed using non-parametric statistical analyses.

BW, BCS, milk production, composition, and fatty acid profiles, as well as lipidic and protein profiles in plasma, were analyzed using repetead measured-analyses of variance (ANOVA). The model included treatment, a period during the experimental phase (Per1 and Per2), and their interaction, whereas it included only treatment as a fixed effect during the pre- and post-experimental periods. A comparison between experimental groups was performed when the treatment or the interaction between treatment and period was significant (*p* < 0.05).

Correlation analyses were performed to evaluate the relationships between the contents of lipoproteins and cholesterol or triglycerides in blood plasma.

The number and size of CL, as well as progesterone level in plasma and pregnancy length, prolificacy, and total reproductive wastage (the difference between the number of CL from pregnant ewe and lambs born/ewe) were analyzed using Student’s *t*-test. Pregnancy rate at day 28, lambing rate, and fetal losses were assessed using Chi-square and/or Fisher’s exact test.

All analyses were performed using the RStudio software. Statistical differences were established at 5% (*p* < 0.05) and results were expressed in terms of means ± standard error and percentages.

## 9. Results

Total concentrate intake was close to the amount fed, being the average DM intake of concentrate 96.90 ± 0.44 and 96.67 ± 0.37% of the offer for CT and LO ewes, respectively. Total estimated net energy (NE_L_) intake was almost identical between CT and LO groups (2.33 ± 0.01 and 2.33 ± 0.01 Mcal/ewe/day; for CT and LO groups, respectively), whereas dietary CP percentage was slightly higher in CT (11.80 ± 0.04%) than LO (11.17 ± 0.05%) due to marginally higher content of CP in the offered diet (Table 1). Total mineral intake was also similar between experimental groups (15.65 ± 3.46 and 15.77 ± 6.05 g/ewe/day; for CT and LO groups, respectively).

BW and BCS changes were undifferentiated between groups across the whole experiment. Additionally, BW and BCS variation between PRE and POST periods were not different between groups (BW: 2.28 ± 0.43 and 2.38 ± 0.31 Kg for CT and LO groups, respectively; BCS: 0.19 ± 0.05 and 0.21 ± 0.05 for CT and LO groups, respectively).

Lipidic and protein profiles in plasma were similar between both CT and LO groups before the start of the experimental period (PRE period; Figure 1).

Nevertheless, there were differences between groups for cholesterol, HDL, and triglycerides during the supplementation period. Despite the lack of differences between groups for LDL, the CT group showed lower concentrations in Per2. Non-esterified fatty acids (NEFAs), total protein, albumin, and urea were not different between groups (Table 2).

Correlations were positive and significant between LDL and triglycerides (r = 0.387; *p* = 0.000), LDL and cholesterol (r = 0.585; *p* = 0.000), and HDL and cholesterol (r = 0.869; *p* = 0.000).

Metabolic profile with reference to lipid and protein was also similar between groups after supplementation (POST period), except for NEFA, which was higher in the CT group (*p* < 0.05; Figure 1).

Milk yield, fat, protein lactose, SCC, and casein levels were not different between CT and LO groups during the supplementation period (Per1 and Per2) (Table 3a). Similarly, differences were not observed before (PRE period) and after experimental supplementation (POST period; Table 3b).

Additionally, fatty acid profiles were similar between groups in the PRE period. However, these differed significantly between LO and CT groups during the supplementation period (Table 4a). The SFA content (in terms of percentage) was lower in the LO group compared to the CT group.

The content of UFA was consistently higher in LO than in CT. MUFAs were also higher in the LO group but only in Per1, while total PUFA was higher in LO in both periods, although the effect was stronger in Per1 (interaction group × period *p* < 0.01). In contrast, PUFA-ω3 showed higher levels in LO but only in Per2 (interaction group x period *p* < 0.01). 

PUFA-ω6 and total trans FA were consistently higher in LO throughout the whole supplementation period.

After the supplementation period (POST), the contents of UFA, MUFA, PUFA, and PUFA-ω6 were still higher in LO groups, while SFA was lower in this, as compared to the CT group.

One ewe from the LO group was eliminated from reproductive analyses due to reproductive problems. Ovulation rate and progesterone circulating levels on days 11 and 18 did not differ between experimental groups in both pregnant and non-pregnant ewes. However, corpora lutea were bigger in pregnant ewes from the LO group compared to CT ones (*p* < 0.01; Table 5). Pregnancy rate at day 28, pregnancy length, lambing rate, and prolificacy did not differ between groups. Additionally, total reproductive wastage and fetal loss rate were statistically similar between groups (Table 6). 

## 10. Discussion

In this trial, diets were designed to be isocaloric and isoproteic between experimental groups. This aim was fully achieved for the energy and partially for the protein, although differences between groups were small. Moreover, the CT and LO diet was formulated to offer a non-fat and fat-enriched supplement, respectively, such as what has been reported in cattle and sheep [17,18]. Since the intake of energy was similar between CT and LO groups, BW, BCS, and milk production were not affected by diets during this trial. These findings are in agreement with different studies carried out in dairy cattle, sheep, and goats supplemented with different forms [3,19,20] and doses of LO [20,21,22,23].

Metabolic status linked to higher cholesterol, triglycerides, and HDL levels found in the LO compared to the CT group can be attributed to the higher fatty acid intake, as already reported in cattle and goats [24,25,26]. In cattle, both LDL and HDL are linked to cholesterol, triglycerides, and phospholipids transport. LDL contributes, together with chylomicrons and HDL, to distribute the triglycerides to peripheral tissues [24,27] having a higher affinity to triglycerides compared with HDL [28]. In our study, the non-supplemented group showed a lower content of triglycerides during the Per2 period, which coincides with a lower content of LDL during the same period. It confirms the affinity between LDL and triglycerides previously reviewed by Bouchart [28]. Additionally, our results demonstrate the relationship between HDL and cholesterol levels. In this sense, the role of HDL consists of mobilizing circulating cholesterol to the liver to be finally metabolized. HDL thus avoids its accumulation in peripheral tissues impacting positively on animal health and welfare [25]. Therefore, HDL increase is probably more related to cholesterol changes than the proper effect of LO.

In the present study, the decrease in NEFA levels and lack of differences between groups during the feeding period confirms the energetic balance between experimental diets. The differences observed between experimental groups in the POST period suggest a residual effect of bypass LO supplementation in limiting fatty acid mobilization from adipose tissue and thus favoring a positive energy balance [29,30]. 

The similar milk fat and protein levels obtained from the CT and LO groups reinforce the energetic and protein balance of experimental diets. This finding is in agreement with those reported by Gonthier et al. [21] in dairy cows and by Gómez-Cortés et al. [31] in ewes fed with flaxseed/linseed supplementation. Our results also showed that milk fatty acid profiles were affected by diets. The increase in milk UFA, MUFA, PUFA, and PUFA-ω3 concentrations demonstrate that bypass LO allowed for absorption at the gut level and distribution in body tissues, including udder tissues [32]. The higher levels of PUFA-ω6 observed in the LO group in our study are attributed to the presence of PUFA (mainly linoleic acid) in the supplement used at a lower concentration than ALA (7.79%; information provided by the manufacturer), which coincides to that reported by Zachut et al. [33] in dairy cattle. Similarly, the higher milk content of SFA in the CT group demonstrates the higher SFA content in the CT diet. Moreover, the higher trans FA concentration in milk samples from the LO group, already found by Cabiddu et al. [18] in dairy sheep supplemented with high levels of a non-protected ALA, is probably due to the partial biohydrogenation of bypass linseed oil at the ruminal level, as suggested by Jenkins and Bridges [9]. However, this pathway was not assessed because it was beyond the scope of this study. Trans human FA are considered putatively noxious to humans [34], although a recent review suggests that trans FA of industrial origin are more noxious than ruminant-sourced trans FA [35]. 

In our study, there was a slight difference in protein content and intake between CT and LO diets. However, the lack of differences between experimental groups, in terms of plasmatic levels of protein profile (total protein, albumin, and urea), demonstrate that CT and LO diets were basically isoproteic. As a consequence, the percentage of milk protein and casein were not affected by the diets. These milk values are in line with those previously reported in dairy cattle [3]. 

In the present study, the large luteal size observed on day 11 (Per1; onset of maternal recognition of pregnancy in sheep [36]) in the LO group coincides with high milk PUFA-ω3 content observed in the same period. In this sense, Petit et al. [37], suggest that PUFA-ω3 promotes granulosa cell proliferation, which generates large luteal tissue and stimulates P_4_ synthesis. Although P_4_ differences were not detected statistically between groups, the LO group showed P_4_ levels were numerically higher compared with the CT group. In this sense, the luteoprotective effect of ALA is linked to PGE_2_ synthesis promoting luteal development and its function [38]. Thus, detailed studies must be performed to evaluate the effect of dietary LO on luteal function, specifically during maternal recognition of the pregnancy period in ewes. 

Pregnancy rates recorded in this study are in line with previous results from our laboratories [39]. Reproductive performances (pregnancy and lambing rates, total reproductive wastages, fetal losses) showed no effect of bypass LO. This lack of effect was also observed in goats, cattle, and sheep [24,30,40]. On the other hand, other studies showed a positive effect of LO to avoid embryo losses in cattle [41]. 

The lack of differences in reproductive performances could be attributed to a “delay effect” of bypass LO on the timing of ovulation. However, the information about this issue is limited. Zachut et al. [33] indicate a longer estrous length compared with the control group, suggesting a delayed timing of ovulation in cows fed with bypass LO. This finding was also reported by Mahala et al. [24] in goats supplemented with FO. Zachut et al. [33] attributed this effect to slow development of preovulatory follicles. The dietary supply of LO used in our study (3.0% DM) falls in the range of the above-quoted studies performed in lactating dairy cattle using bypass LO (2.9% DM [23], 3.8% DM [33]). However, conversely to our experiment, in these studies, the authors used a luteolytic agent to induce the follicular phase. This suggests that additional research is needed to elucidate the effects of LO on ovarian follicular dynamic and timing of ovulation during natural and or induced follicular phase using progestogen and eCG. 

## 11. Conclusions

In conclusion, the results obtained in the present study show that the use of dietary bypass LO modifies the circulating concentrations of lipid metabolites and milk fatty acid profiles, demonstrating the ALA uptake and utilization by body tissues. However, by-pass LO supplementation during the periconceptional period did not increase the reproductive indexes in ewes, contrary to what was reported in previous studies. 

Nevertheless, LO supplementation could be used to increase the PUFA-ω3 of sheep milk supplied to the dairy industry, although the higher level of trans FA could partially counterbalance its beneficial effects on consumer health. 

## Figures and Tables

**Figure 1 animals-13-00280-f001:**
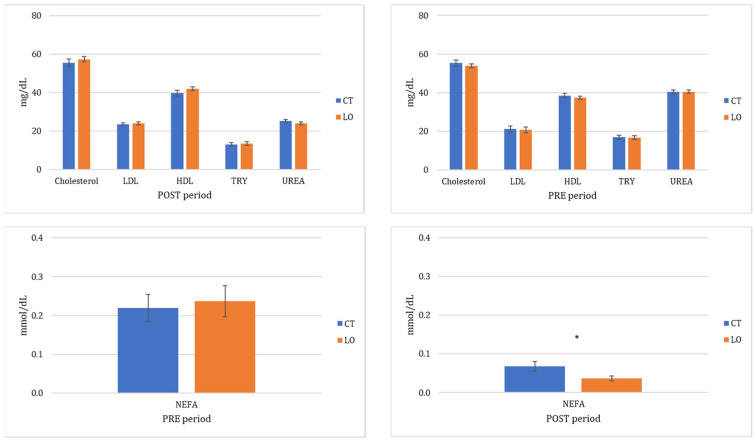
Circulating levels of cholesterol, HDL, LDL triglycerides, NEFAs, total protein, albumin, and urea in Sarda dairy ewes non-supplemented (CT) and supplemented with bypass linseed oil (LO; on PRE and POST periods). An asterisk (*) indicates differences between groups within a period at *p* < 0.05.

**Table 1 animals-13-00280-t001:** Ingredients and chemical composition of diets fed to Sarda dairy ewes: non-supplemented (CT) and supplemented with bypass linseed oil (LO).

Ingredients (g/head)	CT	LO
SILA	-	60.00
Pellets	300.00	200.00
Barley	200.00	120.00
Lucerne hay	600.00	600.00
Ryegrass hay	1000.00	1000.00
Total	2100.00	1980.00
**Total Nutrients (% DM)**	**CT**	**LO**
DM	87.49	87.73
CP	12.32	11.71
NDF	50.18	50.78
ADF	30.06	30.92
ADL	4.65	5.05
EE	2.24	4.52
Starch	6.92	4.62
Ash	8.92	9.21
Total NE_L_ (Mcal/ewe/day)	2.43	2.43

DM: dry matter; CP: crude protein; NDF: neutral detergent fiber; ADF: acid detergent fiber; ADL: acid detergent lignin; EE: ether extract; NE_L_: net energy for milk production.

**Table 2 animals-13-00280-t002:** Circulating levels of cholesterol, HDL, LDL triglycerides, NEFAs, total protein, albumin, and urea in Sarda dairy ewes non-supplemented (CT; *n* = 24) and supplemented with bypass linseed oil (LO; *n* = 24; from Per1 to Per2).

	Per1		Per2		Group (G) Means		*p*-Value		
Metabolites	CT	LO	CT	LO	CT	LO	Per	G	Per × G
Cholesterol (mg/dL)	65.13 ± 2.07	68.27 ± 2.00	51.12 ± 1.71	58.21 ± 1.75	58.12 ± 1.63 ^A^	63.24 ± 1.51 ^B^	0.000	0.008	0.296
LDL (mg/dL)	29.29 ± 0.90 ^a^	27.85 ± 0.98 ^a, c^	21.10 ± 0.73 ^b^	24.44 ± 1.17 ^c, b^	25.19 ± 0.83	26.14 ± 0.79	0.000	0.323	0.014
HDL (mg/dL)	42.78 ± 1.32	46.83 ± 1.21	36.97 ± 1.28	42.48 ± 1.27	39.87 ± 1.00 ^A^	44.66 ± 0.92 ^B^	0.000	0.000	0.566
Triglycerides (mg/dL)	17.24 ± 1.06	17.89 ± 1.04	12.77 ± 0.61	16.27 ± 1.02	15.00 ± 0.66 ^A^	17.08 ± 0.73 ^B^	0.002	0.025	0.135
NEFA (mmol/dL)	0.14 ± 0.01	0.17 ± 0.01	0.09 ± 0.01	0.11 ± 0.01	0.12 ± 0.01	0.14 ± 0.01	0.000	0.122	0.871
Total protein (g/dL)	6.85 ± 0.09	6.76 ± 0.09	6.75 ± 0.10	6.68 ± 0.08	6.80 ± 0.07	6.72 ± 0.06	0.308	0.360	0.955
Albumin (g/dL)	2.74 ± 0.03	2.78 ± 0.02	2.68 ± 0.04	2.73 ± 0.03	2.71 ± 0.02	2.76 ± 0.02	0.902	0.102	0.943
Urea (mg/dL)	24.89 ± 0.83	24.19 ± 0.83	24.48 ± 1.23	22.60 ± 0.75	24.69 ± 0.74	23.39 ± 0.57	0.283	0.167	0.527

Per: period; G: group; Per1: day −1; Per2: day +11. Repeated measures ANOVA. ^a, b, c^ Different lowercase letters indicate differences between groups’ means within the row when the effect of Per × G is at a level of *p* < 0.05. ^A, B^ Different uppercase letters indicate differences between groups’ means.

**Table 3 animals-13-00280-t003:** (**a**)**.** Milk yield and milk composition in Sarda dairy ewes non-supplemented (CT; *n* = 24) and supplemented with bypass linseed oil (LO; *n* = 24) in dairy Sarda ewes non-supplemented (CT) and supplemented with bypass linseed oil (LO; from Per1 to Per2). (**b**)**.** Milk yield and milk composition in Sarda dairy ewes non-supplemented (CT; *n* = 24) and supplemented with bypass linseed oil (LO; *n* = 24) in dairy Sarda ewes non-supplemented (CT) and supplemented with bypass linseed oil (LO) before (PRE) and after (POST) the supplementation period.

(a)									
	Per1		Per2		Group (G) Means		*p*-Value		
Milk	CT	LO	CT	LO	CT	LO	Per	G	Per × G
Yield (g/d)	585.83 ± 43.76	554.78 ± 37.16	590.44 ± 48.40	560.87 ± 47.69	588.09 ± 32.21	545 ± 31.56	0.890	0.358	0.811
Fat (%)	5.45 ± 0.20	5.58 ± 0.15	5.44 ± 0.20	5.75 ± 0.17	5.44 ± 0.14	5.66 ± 0.11	0.656	0.232	0.601
Protein (%)	5.07 ± 0.10	4.908 ± 0.09	5.05 ± 0.12	4.93 ± 0.09	5.06 ± 0.08	4.92 ± 0.07	0.992	0.181	0.821
Lactose (%)	3.67 ± 0.19	3.53 ± 0.19	3.71 ± 0.15	3.67 ± 0.17	3.69 ± 0.12	3.60 ± 0.13	0.811	0.574	0.460
SCC (log n/mL)	2.67 ± 0.15	2.83 ± 0.10	2.51 ± 0.15	2.59 ± 0.12	2.51 ± 0.15	2.59 ± 0.12	0.129	0.356	0.740
Casein (%)	3.76 ± 0.11	3.60 ± 0.1	3.75 ± 0.12	3.66 ± 0.10	3.76 ± 0.08	3.63 ± 0.07	0.784	0.158	0.772
**(b)**									
**Period**		**Milk**		**CT**		**LO**		***p*-Value**	
PRE		Yield (g/d)		920.83 ± 43.76		908.33 ± 61.67		0.898	
POST				465.83 ± 48.58		441.30 ± 40.33		0.701	
PRE		Fat (%)		6.51 ± 0.17		6.61 ± 0.13		0.657	
POST				7.23 ± 0.32		6.89 ± 0.27		0.419	
PRE		Protein (%)		5.38 ± 0.09		50.40 ± 0.09		0.856	
POST				5.44 ± 0.16		5.27 ± 0.10		0.378	
PRE		Lactose (%)		4.29 ± 0.12		4.26 ± 0.08		0.434	
POST				3.53 ± 0.19		3.45 ± 0.17		0.485	
PRE		SCC (log n/mL)		2.39 ± 0.12		2.49 ± 0.09		0.553	
POST				2.79 ± 0.12		2.90 ± 0.14		0.545	
PRE		Casein (%)		4.10 ± 0.08		4.11 ± 0.08		0.955	
POST				4.09 ± 0.16		3.94 ± 0.10		0.421	

(a) Per: period; G: group; Per1: day −1; Per2: day +12. Repeated measures ANOVA. (b) PRE period: day −21; POST period: +20. One-way ANOVA.

**Table 4 animals-13-00280-t004:** (**a**). Milk fatty acid (FA) composition expressed on % fatty acid methyl esters (FAMEs) in Sarda dairy ewes non-supplemented (CT, *n* = 24) and supplemented ewes with bypass linseed oil (LO; *n* = 24; from Per1 to Per2). (**b**). Milk yield and milk composition in dairy Sarda ewes non-supplemented (CT; *n* = 24) and supplemented with bypass linseed oil (LO; *n* = 24) in dairy Sarda ewes non-supplemented (CT) and supplemented with bypass linseed oil (LO) before (PRE) and after (POST) the supplementation period.

(a)									
	Per1		Per2		Group (G) Means		*p*-Value		
Fatty Acids (%) ^1^	CT	LO	CT	LO	CT	LO	Per	G	*p* × G
SFA	70.15 ± 0.74	62.28 ± 0.50	73.42 ± 0.77	68.03 ± 0.70	71.75 ± 0.58 ^A^	65.16 ± 0.60 ^B^	0.000	0.000	0.074
UFA	27.93 ± 1.33	36.00 ± 1.23	26.00 ± 0.89	30.35 ± 1.13	26.98 ± 0.81 ^A^	33.17 ± 0.92 ^B^	0.002	0.000	0.113
MUFA	22.65 ± 0.92 ^a^	29.16 ± 0.71 ^b^	21.03 ± 0.70 ^a^	23.87 ± 0.82 ^a^	21.85 ± 0.59 ^A^	26.52 ± 0.66 ^B^	0.000	0.000	0.023
PUFA	6.95 ± 0.16 ^a^	8.02 ± 0.12 ^b^	4.77 ± 0.13 ^c^	7.64 ± 0.19 ^b^	5.88 ± 0.19 ^A^	7.83 ± 0.11 ^B^	0.000	0.000	0.001
PUFA-ω3	3.11 ± 0.05 ^a^	3.28 ± 0.05 ^a^	1.41 ± 0.04 ^b^	3.12 ± 0.07 ^a^	2.28 ± 0.13 ^A^	3.20 ± 0.04 ^B^	0.000	0.000	0.001
PUFA-ω6	2.70 ± 0.02	3.31 ± 0.07	2.65 ± 0.09	2.97 ± 0.07	2.68 ± 0.06 ^A^	3.14 ± 0.06 ^B^	0.015	0.000	0.074
Trans	2.33 ± 0.17	3.69 ± 0.24	2.16 ± 0.17	3.12 ± 0.25	2.25 ± 0.12 ^A^	3.41 ± 0.18 ^B^	0.082	0.000	0.417
**(b)**									
**Period**		**Milk**		**CT**		**LO**		***p*-Value**	
PRE		SFA (%)		67.59 ± 0.50		67.65 ± 0.53		0.941	
POST				69.91 ± 0.54 ^A^		66.54 ± 0.56 ^B^		0.001	
PRE		UFA (%)		25.86 ± 0.79		26.25 ± 0.75		0.723	
POST				28.60 ± 0.70 ^A^		31.64 ± 1.07 ^B^		0.020	
PRE		MUFA (%)		21.08 ± 0.50		21.61 ± 0.46		0.440	
POST				23.54 ± 0.54 ^A^		26.19 ± 0.71 ^B^		0.005	
PRE		PUFA (%)		7.94 ± 0.17		7.70 ± 0.16		0.314	
POST				5.37 ± 0.11 ^A^		6.11 ± 0.10 ^B^		0.000	
PRE		PUFA-ω3 (%)		1.88 ± 0.07		1.78 ± 0.07		0.315	
POST				1.79 ± 0.06		1.79 ± 0.08		0.997	
PRE		PUFA-ω6 (%)		3.96 ± 0.06		3.95 ± 0.05		0.874	
POST				2.78 ± 0.07 ^A^		3.23 ± 0.05 ^B^		0.001	
PRE		Trans (%)		4.70 ± 0.27		4.52 ± 0.20		0.590	
POST				2.78 ± 0.14		3.23 ± 0.20		0.580	

(a) Per: period; G: group; Per1: day −1; Per2: day +12. Repeated measures ANOVA. ^a, b, c^ Different lowercase letters indicate differences between groups’ means within the row when the effect of Per × G is at a level of *p* < 0.05. ^A, B^ Different uppercase letters indicate differences between groups’ means. ^1^ Fatty acids: SFAs: saturated fatty acids; UFAs: unsaturated fatty acids; MUFAs: monounsaturated fatty acids; PUFAs: polyunsaturated fatty acids; PUFAs-ω3: polyunsaturated fatty acids ω3; PUFAs-ω6: polyunsaturated fatty acids ω6; Trans: trans fatty acid. (b) PRE period: day −21; POST period: +20. One-way ANOVA. ^1^ Fatty acids: SFAs: saturated fatty acids UFAs: unsaturated fatty acids; MUFAs: monounsaturated fatty acids; PUFAs: polyunsaturated fatty acids; PUFAs-ω3: polyunsaturated fatty acids ω3; PUFAs-ω6: polyunsaturated fatty acids ω6; Trans: trans fatty acid. ^A, B^ Different uppercase letters indicate differences between groups’ means.

**Table 5 animals-13-00280-t005:** Ovulation rates, luteal size, and progesterone level in plasma of pregnant Sarda ewes non-supplemented (CT; *n* = 24) and supplemented with bypass linseed oil (LO; *n* = 23).

	CT	LO
	Mean ± S.E.M.	Mean ± S.E.M.
Ovulation rate (Day 11)	1.54 ± 0.12	1.67 ± 0.11
Total area CL/ewe (Day 11; cm^2^)	1.20 ± 0.11 ^a^	1.69 ± 0.12 ^b^
Progesterone on day 11 (ng/mL)	7.87 ± 0.74	8.15 ± 0.98
Progesterone on day 18 (ng/mL)	4.90 ±0.62	4.68 ± 0.43

One-way ANOVA. ^a^ ≠ ^b^: *p* < 0.01.

**Table 6 animals-13-00280-t006:** Fertility indexes after artificial insemination in Sarda dairy ewes non-supplemented (CT) and supplemented with bypass linseed oil (LO).

	CT	LO
	Mean ± S.E.M.	Mean ± S.E.M.
Ewes inseminated (N)	24	23
Pregnancy rate on day 28 (%) *	15/24 (62.50)	11/23 (47.83)
Pregnancy length (Days)	147.57 ± 1.06	149.20 ± 0.42
Lambing rate (%) *	14/24 (58.33)	10/23 (43.48)
Prolificacy (lambs/ewe)	1.36 ± 0.17	1.30 ± 0.15
Total lambs	19	13
Total reproductive wastage **	0.31 ± 0.13	0.50 ± 0.17
Fetal losses (from Day 28; %) ***	1/15 (6.70)	1/11 (9.10)

* Chi-square test, ** number of CL from pregnant ewes—lambs born/ewe, *** Fisher’s test.

## Data Availability

The data presented in this study are available upon request from the corresponding author. The data are not publicly available due to a temporary lack of a publicly accessible repository.

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
