# Peer review of "Effect of Strategic Supplementation of Dietary By-Pass Linseed Oil on Fertility and Milk Quality in Sarda Ewes"

_animals, 2023, doi:10.3390/ani13020280_

Round 1

Reviewer 1 Report

Probably in next study the author may use sheep with a higher milk production and them the probabilty to find differences could be higher. In my opinion you should use a diet with more protein content. (%CP). Probably milk production shoud be affected by sheep breed or diet protein.

Very good study. Congratulations.

Author Response

We would like to thank the referee for his/her time and suggestion for next studies.  In this sense we wish clarify that this trial was performed according to the reproductive management applied in many Mediterranean countries for dairy ewes. The mating season occurs in late spring for adult ewes and, as a result, lactation generally occurs between late autumn and mid-summer. Consequently, ewes are mated/AI when in late lactation. It would be surely interesting to assess the effects of the diets when ewes are in early-mid lactation.

Reviewer 2 Report

The study by Contreras-Solis and colleagues aims at investigating the effects of linseed oil dietary administration on reproductive and productive performances of Sarda ewes.

The subject is of interest although the novelty of the results is quite limited. The experimental design is presented in a very confusing manner, it is not clear the duration of the study (different durations are reported in different sections), the timing of supplement administration, the sampling time, etc. Results are quite confusing and should be presented in a clearer manner, as well.

Simple summary

L23 remove double dot

Abstract

L32-34 describing the method for estrous synchronization is not necessary in the abstract

Materials and methods

L41 how much time after the end?

L96 37 or 38 days? During adaptation, how were animals feed?

L99 Full administration? There was a partial one?

L121 if animal were group fed, how authors are sure about the amount of hay eaten by each single ewe?

L177 blood sampling – - did you had ethical authorization by committee for the protection of animals used for scientific purposes?

L228 this sentence should be inserted in result section

Results

L262 please provide suitable captions for Table 2. Are letters supposed to indicate statistical significance? Please clarify

L302 Table 4a and b, as per Table 2

L319 Ultrasound examination is an operator-dependent procedure. Did the same operator perform all scans on all animals? Was/were the operator/operators blind about the allocation of animals in supplemented and control group? It is interesting that the only parameter that differed significantly among groups was the size of CL, measured subjectively, without a concurrent increase in P4 levels. Please clarify.

Reviewer 3 Report

Dear Authors,

I have revised the manuscript entitled “Effect of Strategic Supplementation of Dietary By-Pass Linseed Oil on Fertility and Milk Quality in Sarda Ewes”. The manuscript is very interesting. Besides, it is well-designed and well-written. However, I have some specific comments:

How green and sustainable is Flaxseed/linseed oil?

Dietary composition:

The authors state that…

“Both diets were designed to supply similar crude protein and energy levels (iso- 123 proteic and isoenergetic diets, (Table 1).”.

However, the EE content in the LO diet is twice higher than that of CT diet. How is it possible that NE was similar to both diets? Could the authors give more details on the energy sources in this diet to explain this difference?

Line 241 does not correlate with Table 1. 
